# An Unsupervised Video Stabilization Algorithm Based on Key Point Detection

**DOI:** 10.3390/e24101326

**Published:** 2022-09-21

**Authors:** Yue Luan, Chunyan Han, Bingran Wang

**Affiliations:** School of Software, Northeastern University (NEU), Shenyang 110169, China

**Keywords:** video stabilization, unsupervised learning, key-point detection, adaptive cropping, RAFT

## Abstract

In recent years, video stabilization has improved significantly in simple scenes, but is not as effective as it could be in complex scenes. In this study, we built an unsupervised video stabilization model. In order to improve the accurate distribution of key points in the full frame, a DNN-based key-point detector was introduced to generate rich key points and optimize the key points and the optical flow in the largest area of the untextured region. Furthermore, for complex scenes with moving foreground targets, we used a foreground and background separation-based approach to obtain unstable motion trajectories, which were then smoothed. For the generated frames, adaptive cropping was conducted to completely remove the black edges while maintaining the maximum detail of the original frame. The results of public benchmark tests showed that this method resulted in less visual distortion than current state-of-the-art video stabilization methods, while retaining greater detail in the original stable frames and completely removing black edges. It also outperformed current stabilization models in terms of both quantitative and operational speed.

## 1. Introduction

With the growing popularity of mobile devices, more and more people are adopting mobile devices to record their lives, but unprofessional shooting often leads to the introduction of shakiness into the final video, which affects people’s subsequent viewing and applications of the video, so more and more people are beginning to pay attention to video stabilization. Especially in complex scenes, the difficulty of video stabilization is increased by the influence of the interactions between the moving foreground and background in the video. It is therefore essential to develop high-performance video stabilization algorithms for shaky video in complex scenes.

The process of a video stabilization algorithm generally starts with the detection of key points, obtaining the unstable motion trajectory according to the key points, then smoothing the motion trajectory, and finally generating stable frames.

Traditional video stabilization methods have the following shortcomings:Key-point detection relies on manual methods, such as ORB and SIFT. However, the detection will be inaccurate or fail when there are drastic changes in external conditions, such as light, darkness, large movements, and the video itself is not of high quality, etc. [1,2,3];Traditional methods of key-point tracking usually use optical flow or Kalman filtering to track the key points. This assumes that there is no significant motion between the two frames and that the grayscale between the two frames is unchanged. However, the scenes in real life are ever-changing, and this produces a large mismatch for complex videos [4];The traditional method of trajectory smoothing uses a fixed kernel of smoothing. However, because the trajectories are ever-changing, the use of a fixed kernel does not achieve the best stabilization effect [2,5,6,7,8,9]. It can also be achieved by fitting to obtain the smoothed trajectories, but this method involves a high levle of error [10];Over-cropping the black edge of the generated frames results in the final frame retaining a smaller field of view and less detail [1,5,7].

DNN-based methods avoid the drawbacks of the traditional algorithm. However, there are still several problems:There are few pairs of stable/unstable videos in the existing official datasets, which is not conducive to supervised models [11,12,13];Most of the existing stabilization models still use traditional methods to detect key points in the key-point detection stage, but the traditional methods do not detect the key points completely and many key points are lost. In the key-point detection stage of the video stabilization method using a DNN-based model, although the key points generated are abundant, the method does not consider the impact of redundant key points in a large untextured area [14];For generated frames, a fixed cropping method is used to remove black edges, which preserves less of the original frame content [11,12,13,14,15,16].

In this paper, an unsupervised video stabilization algorithm based on key-point detection is proposed to deal with the above problems. A DNN-based key-point detector was first used to detect key points. The key points and optical flow inside the untextured area with the largest area were optimized to obtain an accurate unstable motion trajectory. The motion trajectory was then placed into the smoother to adaptively generate a smoothing kernel to smooth the motion trajectory iteratively. Finally, based on the smoothed trajectory, a homography transformation was applied to generate a stable frame with black edges. For the processing of the black edges, in this study we used an adaptive cropping method to crop and retain the maximum area of the original frame.

The four contributions of this paper are as follows:An unsupervised deep learning model is used for video stabilization, which does not require pairs of stable/unstable datasets;A method is proposed to optimize key points and optical flow in large texture-free areas to obtain accurate non-stationary motion trajectories;Adaptive cropping is proposed to retain the maximum field of view for the stable video, while not having a black edge; andThe model runs faster on the basis of improved model performance, which is more suitable for the requirements of video stabilization.

## 2. Related Work

In this section, we summarize traditional stabilization methods and DNN-based stabilization methods. Traditional stabilization methods include 2D and 3D methods.

**2D methods.** They use 2D transforms to describe the 2D motion trajectory in the video and stabilize the video based on these transforms. For example, MeshFlow [1] uses a traditional method for key-point detection, then uses Kalman filtering [17] to track key points and obtains motion trajectories. The motion trajectories are smoothed by optimizing the minimum energy function and stable frames are generated by means of grid-based warping. Liu et al. optimized camera trajectories by means of bundling to achieve motion smoothing, and generated stable frames based on the grid [5]. Wang et al. developed a stabilizer based on finding the relative position of key points [6]. The developers of the Steadyflow [7] method used Jacobi’s method to smooth the motion trajectories. Matsushita Y et al. achieved video stabilization by interpolating and inpainting the video [8]. Grundmann M et al. achieved video stabilization by optimizing the camera path using L1 [9]. Hu R et al. used SIFT for keypoint detection, and these key points were used to estimate the motion trajectories of the camera [3]. The 2D method is relatively simple to implement, but its time complexity is high. However, when it comes to complex scenes, the method’s processing capability for stabilizing video is relatively limited. This is because the focus when stabilizing a simple scene is only the jitter factor introduced by the camera, whereas in a complex scene one also needs to consider whether there is a moving foreground target in the video. At the same time, the 2D method only uses a fixed smooth kernel in the process of generating stable frames, but unstable trajectories are various, so it is incorrect to use a fixed smooth kernel.

**3D methods.** They find the camera trajectory in 3D space and smooth it to achieve video stabilization. Liu and Wang et al. used a depth camera for stabilization [2]. Liu and Gleicher et al. projected the 3D feature points onto a new camera location for stabilization [10]. Karpenko et al. used a gyroscope sensor for stabilization [18]. Zhang et al. stabilized video sequences based on a 3D perspective camera model [19]. Bhat P et al. modified the multi-view stereo algorithm to calculate the view-dependent depth for video stabilization [20]. 3D methods obtain more information and therefore perform better. However, the 3D stabilization algorithm has strict hardware conditions and limited applications.

**DNN-based stabilization methods.** They aim at the transformation of unstable frames to stable frames, whereas traditional stabilization methods aim at the obtainment of motion trajectories. Yu and Ramamoorthi et al. achieved the generation of stable frames by inputting the processed accurate optical flow into the network to generate the warping fields of stable frames [15]. The advantage of this method is that the optical flow is accurate. The disadvantage is that the post-processing of the optical flow leads to complex operation costs and long running time. Furthermore, optical flow over-patching can result in local distortion. Moreover, this method uses a manual key-point detector, which may fail to detect complex video or texture-free region cases. Xu et al. obtained the stable video directly using unsupervised learning models [14]. The advantages of this approach are its short running time, clear model logic, and the lack of a need for training data pairs. However, the disadvantage is that the key points are generated using a depth model, so redundant key points will be generated, especially in regions with relatively large untextured areas. Furthermore, the motion trend inside the large untextured area is different from the overall motion trend. This means that the final motion trajectories obtained are inaccurate, which introduces distortion. Wang et al. obtained the homography matrix of the grid through the use of a Siamese network, which generated stable frames [11]. This method has the advantage of using both unstabilized frames and frames that have been stabilized up to the current time as inputs to obtain the warping field until all stabilized frames are generated, which is performed cyclically, as it does not stabilize only one frame individually. The disadvantages of this method are that, firstly, many training data pairs are required, and secondly, the model only uses the simple homography matrix to generate stable frames, so the method will not work for videos with violent motion and close-up scenes. Choi et al. used interpolation for video stabilization but introduced artifacts [21]. The advantage of this approach is that the full-frame is stabilized, so there are no cropping problems with the final stabilized video obtained. The disadvantage is that the generated frames can introduce artifacts due to the need for multiple interpolations. The PWStableNet method [12] introduces a cascade structure to generate pixel-by-pixel warp fields for video stabilization. The advantage of this method is that it takes into account the fact that different pixels may have different distortions, so it stabilizes in a pixel-by-pixel manner in adjacent frames, and the use of a cascade structure is beneficial to learning the warp field more accurately. A disadvantage of this method is that it uses pixel-by-pixel stabilization, so the running time is relatively long, and it also requires a large number of training data pairs. It is not as effective as the stable grid-based model for complex video processing.

In contrast, the method proposed in this paper uses an unsupervised DNN-based model for video stabilization, so the requirements on the dataset are not strict. For videos of complex scenes, we focus on the problem of obtaining rich key points, while optimizing key points and optical flow over large untextured areas to generate more accurate motion trajectories. In this study, we used SuperPoint [22] to generate rich key points, then used gradient information to optimize key points and optical flow so that the final result would have less distortion. For the problem of black edges in the generated frames, an adaptive cropping method was used to ensure that the black edges were completely removed, while retaining more of the original information.

## 3. Pipeline

The overall structure of the algorithm is shown in Figure 1. The algorithm is divided into four stages: the coarse key-point detection and tracking stage, the accurate key-point detection and accurate optical flow stage, the smooth motion trajectory stage, the stable frame generation and adaptive cropping stage.The above stages are described in Section 3.1, Section 3.2, Section 3.3 and Section 3.4.

### 3.1. Coarse KeyPoint Detection and Tracking Stage

This stage is divided into two parts, one involves using SuperPoint to detect key points and return the matched key points, and the other involves using RAFT [23] to generate the optical flow iteratively. The stage is illustrated in Figure 2. This optical flow is then combined with the key points returned by SuperPoint to complete the tracking of the key points.

In the optical flow stage, we refer to the RAFT method [23] developed by Zachary Tee et al., whereas in the key-point detection stage, we refer to the SuperPoint method of Daniel Detone et al. [22].

This stage takes adjacent frames $I_{i}$ and $I_{i+1}$, *i*∈{0,C}, where C is the total number of frames. It uses SuperPoint for key-point detection to obtain $kp_{i}$ and $kp_{i+1}$, which are matched to obtain the matched pairs $\left\{m_{i},m_{i+1}\right\}$. Then $I_{i}$ and $I_{i+1}$ are input into RAFT to obtain the optical flow $F_{i}$. According to $F_{i}$, the motion vector $f_{i}$ of $m_{i}$, at the position corresponding to the key point in $F_{i}$, is extracted. $m_{i}$ is added to the motion vector $f_{i}$ to obtain the position of the $m_{i}$ in relation to the next frame, which completes the tracking of the key points and obtains the motion trajectories of the key points. As shown in Figure 3, the reason for adopting SuperPoint was that SuperPoint is robust compared with manual key-point detection methods. Manual methods may miss many potential points of interest. Another problem is that SuperPoint produces too many key points, as shown in Figure 4. As a result, for videos with large untextured areas, redundant key points will be generated and distortion will be introduced. This problem is solved in the second stage of this paper.

### 3.2. Accurate Alignment and Motion Trajectory Stage

The second stage mainly involves obtaining accurate key points and optical flow, as well as obtaining the trajectory of the keypoints. This stage is divided into two parts. One is related to obtaining the accurate alignment, and the other is related to obtaining the motion trajectories. The accurate alignment portion of this stage includes the generation of the mask, ensuring the precision of the key points and the precision of the optical flow, as shown in Figure 5.

#### 3.2.1. Mask Generation

From experiments, we know that key-point redundancy in large untextured areas and inaccurate optical flow inside large untextured areas may distort the generated stable frames. Thus, in our method, we obtain the mask of the *i*-th frame by means of a gradient, expressed as $M_{i}$: $M_{i}^{\left(x,y\right)}$ = 0, where $G_{i}^{\left(x,y\right)}$≥ 3$\bar{G_{i}}$, (x,y) denotes the position of the pixel and $\bar{G_{i}}$ denotes the average of the gradient of the *i*-th frame.

The visualization of $M_{i}$ according to the gradient, denoted by $part_{1}$ in Figure 5, shows that there are many burrs around the untextured area and irregularities, which will affect the subsequent processing. Therefore, the operation of dilation and erosion on $M_{i}$ is used to obtain $M_{i}^{{}^{\prime }}$.

Connecting small and non-continuous non-flat areas at small intervals via dilation, Equation (Equation 1) is as follows: (1)$$\;\mathrm{dilated}\;\left(x,y\right)=\max\left\{M\left(\left(x+x^{\prime },y+y^{\prime }\right)\right)\right\}$$
and rejecting non-flat areas that are too small via erosion, Equation (Equation 2) is as follows: (2)$$\mathrm{eroded}\left(x,y\right)=\min\left\{M\left(\left(x+x^{\prime },y+y^{\prime }\right)\right)\right\}$$
where the area around (x,y) is denoted by $\left(x+x^{{}^{\prime }},y+y^{{}^{\prime }}\right)$$\left(|x^{{}^{\prime }}|\leq 1,|y^{{}^{\prime }}|\leq 1\right)$.

#### 3.2.2. Accurate Key Points

The mask $M_{i}^{{}^{\prime }}$ can be obtained from the mask generation stage. As shown in $part_{2}$ of Figure 5, the maximum untextured area in $M_{i}^{{}^{\prime }}$ is obtained via calculation. We remove the key points within the maximum untextured area to obtain the accurate key points, as shown in Figure 6.

#### 3.2.3. Accurate Optical Flow

The unprocessed optical flow is shown in Figure 7a. It can be noted that the optical flow within the untextured area is not consistent with that in the textured area, which will distort the generated frame in the local region. Next, PCAFlow [24] is used to fill in the optical flow in the removed area based on the optical flow in the unremoved area, so that the motion of the frame tends to be consistent and an accurate optical flow is obtained, as shown in Figure 7b.

#### 3.2.4. Trajectory Generation

The final step is to obtain a mesh-based vertex motion trajectory. Due to the various levels of complexity of the video, in this study we used homography based on the separation of foreground and background and mesh-based vertex profiles to obtain the motion trajectories. In this study, the frames were divided into one grid every 16 pixels, with a total of A × B grids. According to $d_{i}$, the $m_{i}$ value is then propagated to the nearby grid vertices $v_{i}$ according to Equation (Equation 3): (3)$$d_{i}=\sqrt{\left|m_{i}^{2}-v_{i}^{2}\right|}$$
where $m_{i}$ denotes the position of the key points, $v_{i}$ is the position of the mesh vertexes, and “| |” denotes an absolute value.

The next two median filters are used for the motion trajectories obtained for each vertex. The first one aims at assigning each vertex a most representative motion vector and the second one aims at preventing the appearance of noise. The motion vector $f_{i}$ is then divided into two categories representing the background and the foreground of motion according to Equation (Equation 4):(4)$$B,P=\mathrm{Kmeans}\left(f_{i},2\right).$$

According to the above, $\left\{m_{i},m_{i+1}\right\}$ are used in RANSAC to obtain two homographies, $H_{b}$ and $H_{p}$. The local residual motion vectors for the foreground and background are obtained according to Equation (Equation 5): (5)$$\begin{array}{c} {\tilde{v}}_{i}^{p}=m_{i}^{p}-H_{p}\times m_{i+1}^{p}\\ {\tilde{v}}_{i}^{b}=m_{i}^{b}-H_{b}\times m_{i+1}^{b} \end{array}$$
where × means matrix multiplication.

According to Equation (Equation 6), the foreground and background motions are obtained: (6)$$\begin{array}{cc} v^{p} & =v_{i}+{\tilde{v}}_{i}^{p}\\ v^{b} & =v_{i}+{\tilde{v}}_{i}^{b}\\ v & =v^{p}+v^{b} \end{array}$$

### 3.3. Motion Trajectory Smoothing

The unstable trajectory is input into the encoder/decoder structure to generate a smooth kernel for the trajectory, and then the smooth kernel is applied to the iterative smoothing of the unstable trajectories, and finally, the stable trajectories are obtained.

The overall loss function is composed of three parts: temporal smoothness loss $L_{t}$, shape-preserving loss $L_{s}$, and content-preserving loss $L_{c}$:

According to Equation (Equation 7), $L_{t}$ is formed by minimizing the energy function: (7)$$\begin{array}{c} L_{t}=\sum_{t=0}^{T-1}{\left({∥P\left(t\right)-C\left(t\right)∥}_{2}\right)}^{2}+\lambda \sum_{r\in 16}\left({∥P\left(t\right)-P\left(r\right)∥}^{2}\right)\\ C\left(t\right)=\sum_{t=0}^{T-1}v\left(t\right) \end{array}$$
where C(t) denotes the camera path, t denotes time moment, P(t) denotes the optimized path, *r* denotes the radius (*r*∈ 16), and v(t) denotes the motion vector of the mesh vertices.

Suppose that $\left\{p,\hat{p}\right\}$ is the matching pair of key points from $I_{i}$ to $I_{i+1}$, and *p* consists of a closed grid including four vertices, according to Equation (Equation 8): (8)$$\begin{array}{c} p=V_{p}\times w_{p}^{T}\\ V_{p}=\left[v_{ia}{}^{1},\;v_{ia}{}^{2},\;v_{ia}{}^{3},\;v_{ia}{}^{4}\right]\\ w_{p}=\left[w_{p}^{1},\;w_{p}^{2},\;w_{p}^{3},\;w_{p}^{4}\right] \end{array}$$
where $w_{p}$ denotes the bilinear interpolation operation, *a* denotes the grid, and *i* denotes the *i*-th frame.

$\hat{p}$ can then be expressed using $\left[{\hat{v}}_{ia}^{1},{\hat{v}}_{ia}^{2},{\hat{v}}_{ia}^{3},{\hat{v}}_{ia}^{4}\right]$, then the shape-preserving loss $L_{s}$ is built on each grid, according to Equation (Equation 9): (9)$$L_{s}=\sum_{i=0}^{C-1}\sum_{a=1}^{\left(A-1)(B-1\right)}{∥{\hat{v}}_{ia}^{3}-\left({\hat{v}}_{ia}^{2}+R_{90}\left({\hat{v}}_{ia}^{1}-{\hat{v}}_{ia}^{2}\right)\right)∥}_{2}^{2}$$
where $R_{90}=\left[\begin{array}{cc} 0 & 1\\ -1 & 0 \end{array}\right]$.

To prevent distortion around key points, and to avoid distortion between flat and non-flat areas, content-preserving loss $L_{c}$ is added here, according to Equation (Equation 10): (10)$$L_{c}=\sum_{x=0}^{H-1}\sum_{y=0}^{W}{∥p-H_{g}\left(p\right)∥}^{2}$$
where $H_{g}$ is the homography matrix of the grid containing *p*, and *p* is found as in Equation (Equation 8).

### 3.4. Adaptive Cropping

The fourth stage is the adaptive cropping stage. The stable frames generated in the previous stage have black edges. Here, we present an adaptive cropping method designed to crop the stable frames with black edges, cropping all the black edges while retaining more original frames.

The process of this stage first involves obtaining the binary image of the black edge of each frame according to the generated stable frame with a black edge. Then, the binary image is eroded. Finally, the global maximum inscribed rectangle is obtained according to the maximum inner contour of the black edge, and the video frame is cropped according to the inscribed rectangle. The overall processes are as follows.

**Step1:** Find the black-edge binary image, as shown in Figure 8. A specific binary image with only black edges is first obtained from the generated stable frames with black edges. Using a grid-based format, each grid generates its own homography matrix $H_{g}$ based on the motion vectors in the *x* and *y* directions. The original grid vertices generate the target-stabilized grid based on $H_{g}$, which ultimately synthesizes the stable frames. Only the coordinates within *0-H* and *0-W* are the RGB values of the original frames. All other locations greater than *W* and *H*, and less than 0, are black edge locations, from which a specific binary image can be calculated.

**Step2:** The “binary image ” obtained from $step_{1}$ is eroded according to Equation (Equation 2).

**Step3:** Contouring the white area of the “binary image”. This step involves finding the connected domain in the “binary image” with a pixel value of 255, and the same idea is used in the work of Satoshi et al. [25]. In a binary image, when the pixel values between adjacent pixels are different for the first time, this isconsidered to be the start of the boundary $i_{s}$ and is uniquely encoded to deal with the problem of creating multiple boundaries. When you find the the start-point $i_{s}$, then you look for the end-point $i_{e}$, which is used to determine whether the contour has ended.When the current point is found to belong to the contour, it is currently the center point $i_{c}$. In the process of tracking, we look for the next point belonging to this contour in the 8-neighbourhood of the center point, which is the latest point, $i_{n}$. $i_{c}$ and $i_{n}$ are continuously iterated and replaced to complete the generation of the boundary, as shown in Figure 9.

**Step4:** The process of finding the maximum inner rectangle for the contour. The outline of the white region and the set of points of that outline can be obtained from $step_{3}$. Then, according to the characteristics of the rectangle, we define the sum value of the coordinates of the upper left corner of the rectangle as the minimum, and the sum value of the coordinates of the lower right corner as the maximum. Using this rule, we find the coordinates of the two points diagonal to the rectangle using point set of the outline, so we can find the maximum inner rectangle of the outline.

The advantage of adaptive cropping is the ability to avoid the problem of the black edge in generating stable frames and to retain the maximum field of view of the original frames.

## 4. Experiments

We evaluated our model on the public benchmark dataset, the NUS dataset [5], and compared it with the current state-of-the-art models, both DNN-based methods and traditional methods. In this section we provide comprehensive quantitative and qualitative assessment results. Furthermore, during processing, it was found that most models could not handle high-resolution video. The model presented in this paper can handle video of any resolution adaptively.

### 4.1. Datasets

The NUS dataset includes a total of 174 unstable videos of varying duration (10–60 s), which are divided into six categories: Regular, Zooming, Running, QuickRotation, Crowd, and Parallax. The resolutions in the dataset are 640 × 360, 640 × 480, and 1280 × 720.

### 4.2. Experimental Settings

Since the experimental model in this paper was an unsupervised model, there was no need for a training dataset. We just input the unstable video data, implemented the model, and output stable video. The test set used for the model was obtained from the NUS dataset. We ran the model for this paper on a Tesla V100-SXM2 with a 32G GPU.

### 4.3. Subjective Results

The visual results for Meshflow [1], DUT [14], StabNet [16], DIFRINT [21], Yu et al. [15], and the model presented in this paper are shown in Figure 10.

The results from the visualization can be decsribed as follows.

**Video1:** Large untextured area (sky in the video). The MeshFlow [1] and Yu et al. [15] methods produced redundant black edges due to improper cropping. DUT [14] and Yu et al. [15] did not take into account the key points in the untextured area and the optical flow’s inconsistency with the overall optical flow, resulting in significant distortion of the tower in the stable frame, as well as around the buildings. StabNet [16] excessively cropped the image and at the same time introduced more distortion as this method only handles regular two-dimensional motion.

**Video2:** Motion foreground. MeshFlow [1] produced too few key points, resulting in distortion in and around the moving object, as well as excess black edges. DUT [14] and Yu et al. [15] both produced excessive black edges due to improper cropping. DIFRINT [21] produced artifacts around the moving foreground target. DIFRINT [21] resulted in artifacts due to the use of multiple interpolations. StabNet [16] left only a small field of view due to excessive cropping.

**Video3:** QuickRotation. MeshFlow [1], DUT [14], and Yu et al. [15] all produced redundant black edges due to improper cropping. MeshFlow [1] did not adapt to fast rotation using only homography transforms and out-of-place warping. The DIFRINT [21] method uses interpolation and does not adapt to fast-moving objects—neither the camera itself nor the objects in the video. The motion trajectory of the feature points of this type of video is complex, so StabNet [16] did not adapt well and this led to distortion.

**Video4:** Crowd. MeshFlow [1] and Yu et al. [15] produced excessive black edges due to improper cropping. MeshFlow [1] produced distortion, resulting from inaccurate estimation due to the use of homography transformations alone, whereas the results of the method of Yu et al. [15] showed distortion due to the inaccurate optical flow at the filled edges. DIFRINT [21] still produced artifacts due to interpolation. The results produced by StabNet [16] were still distorted due to the fact that this method is not able to adapt well to complex motion.

### 4.4. Quantitative Results

#### 4.4.1. Evaluation Metrics

To fairly assess the quality of the model, three evaluation metrics are introduced here: the cropping ratio, distortion ratio, and stability score.

**The cropping ratio** is the area of the generated frame remaining after the black border has been completely removed. The larger the ratio, the better. The calculation is based on the homography matrix **M**, obtained by matching the key points of the original and stable frames. The cropping ratio **cp** of each frame is obtained from the scale component of **M**. The final crop ratio is equal to the average of all **cp** values, according to Equation (Equation 11): (11)$$\begin{array}{c} \mathrm{cp}=1/\sqrt{M{\left[0,1\right]}^{2}+M{\left[0,0\right]}^{2}}\\ \mathrm{cropping}\_\mathrm{ratio}=\frac{1}{C}\sum_{c=1}^{C}\left(\mathrm{cp}\right) \end{array}$$

The **distortion ratio** is the degree of distortion of the stable frame. A higher value means that a better shape of the original frame has been maintained. The distortion rate is equal to the minimum value of the ratio of the two smallest eigenvalues $\lambda_{1}$ and $\lambda_{2}$ among all the eigenvalues of **M** in all frames, and the distortion fraction is equal to the minimum of all $\lambda_{1}$$\lambda_{2}$ ratios, according to Equation (Equation 12): (12)$$\begin{array}{c} \lambda_{1},\lambda_{2},\lambda_{3},\ldots \ldots ,\lambda_{n}=\mathrm{linalg}.\mathrm{eig}\left(M\right)\\ \lambda_{1},\lambda_{2}=\min\left(\left(\lambda_{1},\lambda_{2},\lambda_{3},\ldots \ldots ,\lambda_{n}\right),2\right)\\ \mathrm{Distation}\_\mathrm{ratio}=\min\left(\frac{\lambda_{1}}{\lambda_{2}}\right) \end{array}$$
where linalg.eig() represents the eigenvalue operation.

**The stability score** is the degree of smoothness of the stabilized video—the larger the better. We select the average of the percentage of translational components and the average of the percentage of rotational components over the frame, the minimum of both is called stability score, according to Equation (Equation 13):(13)$$\mathrm{stability}\_\mathrm{score}=\min\left(\frac{1}{C}\sum_{c=1}^{c}fft\left(\;\mathrm{Trans}\;\right),\frac{1}{C}\sum_{c=1}^{c}fft\left(\;\mathrm{Theta}\;\right)\right)$$
where Theta denotes the rotational component, Trans denotes the translational component, and *fft*() denotes the Fourier transform function.

#### 4.4.2. Quantitative Results

We evaluated our model and state-of-the-art video stabilization methods on the public benchmark dataset NUS, where the models being compared were Meshflow [1], Liu et al. [5], StabNet [16], and DUT [14], Yu et al. [15]. The results of the evaluation of these quality metrics in the demonstrated approach are shown in Table 1, Table 2 and Table 3.

Table 1 shows a comparison of the cropping ratio. In the NUS dataset, the method presented in this paper maintained a similar cropping ratio to that obtained using the methods of Liu et al. [5] and Yu et al. [15]. Meanwhile, compared with the methods of Meshflow [1], DUT [14] and StabNet [16] based on the “one size fits all” cropping method, the cropping ratio obtained using the method presented in this paper has already improved upon the cropping ratio of many other methods.

Table 2 shows a comparison of the distortion ratio. Our model, using the NUS dataset, resulted in less distortion. Compared with MeshFlow’s transform based on global homography [1], the transform based on the foreground and background separation introduced less distortion. Compared with DUT [14], which also uses a DNN-based key-point detector, the problem of key-point redundancy caused by a large untextured area was considered in our experimental process, and the key points in a large untextured area were removed. Furthermore, PCAFlow was used to fit the optical flow, which introduced less distortion. Compared to StabNet [16] which also generates stable frames based on grids, StabNet [16] was not unsupervised and DeepSta datasets were used in the training. This dataset is too small and contains a single scene, so the training was not accurate enough. Compared with the method of Yu et al. [15], which also filled the flat area, Yu et al. [15] filled the boundary between the untextured area and the textured area for a second time, which was overfilled and also produced deformation and distortion. When the object in the video was too close to the camera or had a large occlusion or there were too many people in the video, the motion trajectory optimization method of Liu et al. [5] did not follow its motion trend, resulting in poor results, including distortion and artifacts.

Table 3 shows the stability score obtained in our experiments. In the NUS dataset, our model obtained a high stability score compared to those of the other models. Our model was the best on the whole, especially in Running, Zooming, QuickRotation, and Parallax categories, and in Regular and Crowd the method was ranked second. Our model performed better than Meshflow [1], StabNet [16], and DUT [14], which are also based on the use of a grid. Compared with DUT [14], which also adopts unsupervised learning, the stability score of our model exceeded that of DUT [14] on the NUS datasets. Compared with the method of Liu et al. [5], in regard to smoothing the path of the bundled camera, our model showed superior performance. Compared to the model of Yu et al. [15], which uses optical flow as its input, our model showed superior performance in the Running, Zooming, QuickRotation, and Parallax categories. However, in the Crowd and Regular categories, our model performed slightly worse than that of Yu et al. [15], because, for some videos, the flat area was complicated. For example, when the flat area in the video was very fragmented, our model did not deal with such videos well.

The data in Table 3 are the average values of different models obtained for different classes. If videos were compared on a one-to-one basis, there would be videos with large gaps in their stability scores in each category for the different models. Because the stability score of the gap is relatively small, it is generally not obvious in the subjective results. The stability score is related to many factors, including video content, video brightness and darkness, etc. Thus, if the difference between the stability scores obtained for a single video is small, this is not visually obvious, but if the difference between the stability scores of a video is large, it is visually obvious. So the average represents the average situation of a certain category.

### 4.5. Comparison of Runtimes

Table 4 shows a comparison of the different state-of-the-art methods in terms of their runtime, including traditional and DNN methods. Under the same experimental conditions, the runtime of the model presented in this paper was the shortest compared to the other models. Our model includes fewer operations, so it has a lower runtime. It was found through our experiments that the most time-consuming part of the process the stage involving the generation of the trajectory and obtaining the optical flow. During the trajectory generation process, each key point should be processed separately, and the key points should be propagated to the grid vertices and processed on the grid vertices. The overall time required was as follows:

Grid-based time consumption can be expressed as in Equation (Equation 14):(14)$$T_{g}=A\times B\times C\times n$$
where *n* is the number of key points.

Pixel-based time consumption can be expressed as in Equation (Equation 15):(15)$$T_{p}=H\times W\times C\times n$$
where ***H*** and ***W*** are the resolutions of the videos.

In the trajectory generation stage, we constructed a Boolean matrix by matricizing all operations so that the index could be used to determine whether the current point was required and to extract the corresponding movement deviation based on the index. The extraction of the optical flow was also found to be a time-consuming step when using RAFT. RAFT itself was found to be a means of generating optical flow via iteration, The accuracy of optical flow had little influence on the experimental results between 20 and 10 iterations, so the number of iterations was adjusted to 10, as shown in Figure 11.

## 5. Conclusions

In this paper we present a method for video stabilization using an unsupervised approach. The accurate motion trajectory of key points was obtained by post-processing the key points and the optical flow in the maximum flat area. The trajectory was also smoothed using a deep learning approach. Finally, for the generated frames with black edges, an adaptive cropping method was used to crop them so that the original frame details were preserved as much as possible. On the benchmark dataset, we verified the superiority of our model over other state-of-the-art methods, finding that it had less distortion and larger stability scores, minimizing black edges while maximizing the preservation of the original frames. In terms of computational efficiency, our method was more efficient than the other methods tested.

The model presented in this paper is not yet an end-to-end model, so we hope to improve it into an end-to-end model. Moreover, to further improve the performance of the model presented in this paper, we hope that online stabilization can be implemented.

## Figures and Tables

**Figure 1 entropy-24-01326-f001:**
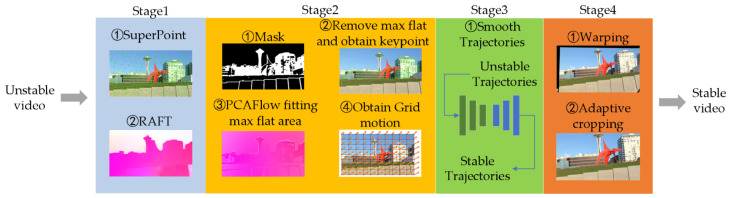
Overall model of the pipeline.

**Figure 2 entropy-24-01326-f002:**
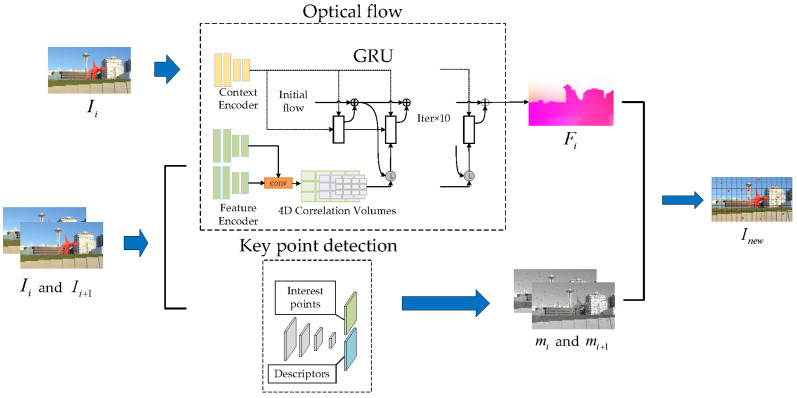
The process of coarse key point detection and tracking stage.

**Figure 3 entropy-24-01326-f003:**
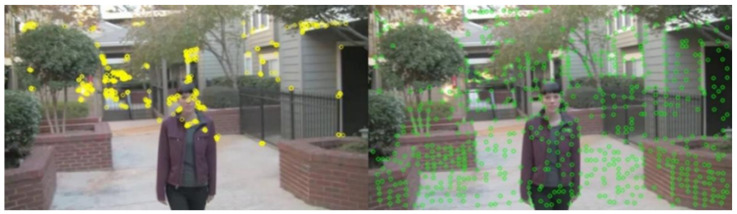
Complex video (moving foreground): the left shows the distribution of key points obtained using the traditional manual method and the right shows the distribution of key points obtained using SuperPoint.

**Figure 4 entropy-24-01326-f004:**
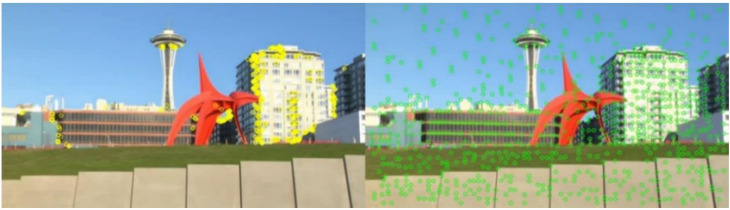
Video of a large flat area (sky): the left shows the distribution of key points generated using manual methods and the right shows the distribution of key points generated using SuperPoint.

**Figure 5 entropy-24-01326-f005:**
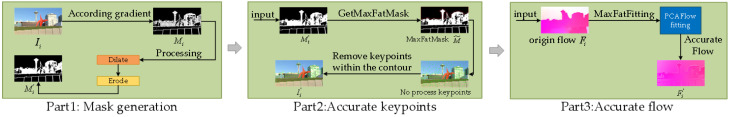
Overall flowchart of the accurate alignment part of the process.

**Figure 6 entropy-24-01326-f006:**
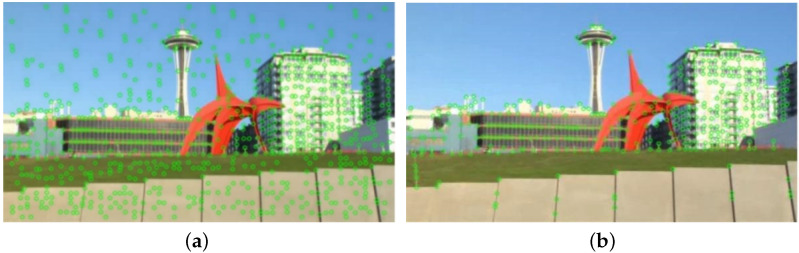
Accurate key points: (**a**) shows key-point detection using SuperPoint only and (**b**) shows the distribution of key points after removing the flat areas.

**Figure 7 entropy-24-01326-f007:**
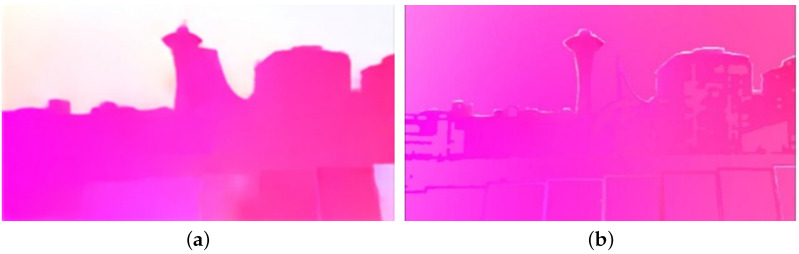
Accurate optical flow: (**a**) shows the optical flow obtained using RAFT for the original frame, (**b**) shows the optical flow obtained by fitting the optical flow to the removed position using PCAFlow.

**Figure 8 entropy-24-01326-f008:**
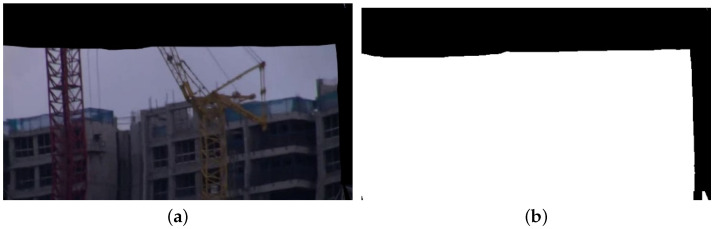
Special binary images for the generated frames: (**a**) shows the generated stable frame with black edges and (**b**) shows the special binary map obtained according to (**a**).

**Figure 9 entropy-24-01326-f009:**
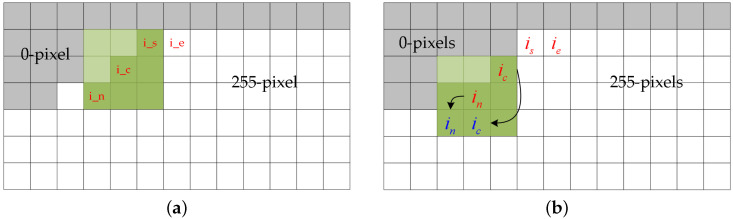
Diagram of the contour tracking process: where $i_{s}$ denotes the starting point, $i_{e}$ the endpoint, $i_{c}$ the central point, and $i_{n}$ the latest point. (**a**) shows the distribution of the points in the current state. (**b**) shows the result of the first trace.

**Figure 10 entropy-24-01326-f010:**
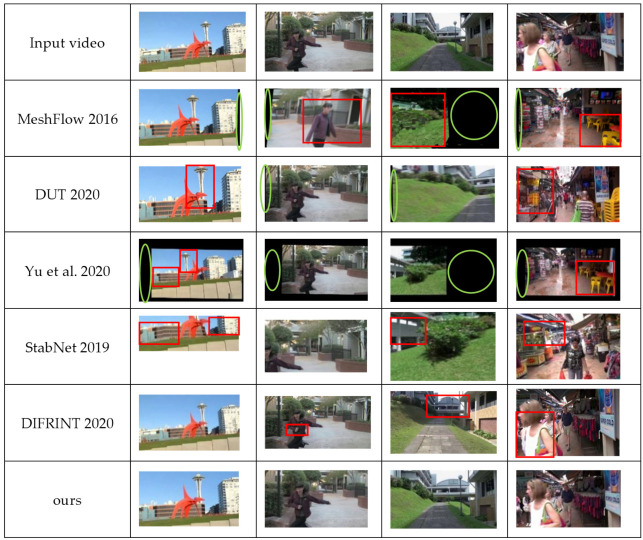
Visual results of different methods, including Meshflow [1], DUT [14], StabNet [16], DIFRINT [21], Yu et al. [15] and the method in this paper. Red boxes indicate distortion, and green circles indicate the presence of black edges.

**Figure 11 entropy-24-01326-f011:**
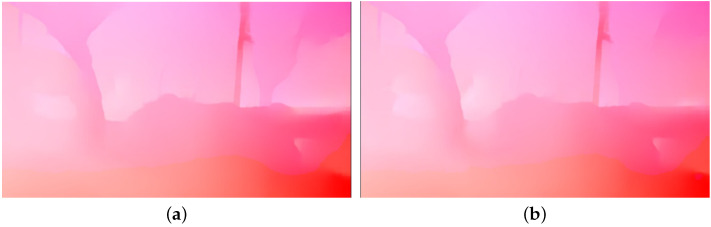
RAFT visualization results for different iterations: (**a**) shows the visualization for 20 iterations, (**b**) shows the visualization for 10 iterations.

**Table 1 entropy-24-01326-t001:** Comparison of cropping ratio.

	Meshflow [1]	Liu et al. [5]	StabNet [16]	DUT [14]	Yu et al. [15]	Ours
Regular	0.69	0.82	0.54	0.74	0.88	0.88
Running	0.59	0.79	0.51	0.69	0.70	0.70
Zooming	0.60	0.79	0.52	--	0.77	0.75
Crowd	0.55	0.82	0.50	0.71	0.82	0.80
QuickRotation	0.38	0.84	0.42	0.67	0.84	0.70
Parallax	0.54	0.83	0.50	0.71	0.82	0.78

**Table 2 entropy-24-01326-t002:** Comparison of distortion ratio.

	Meshflow [1]	Liu et al. [5]	StabNet [16]	DUT [14]	Yu et al. [15]	Ours
Regular	0.90	0.92	0.70	0.98	0.97	0.98
Running	0.76	0.83	0.75	0.84	0.90	0.92
Zooming	0.80	0.74	0.72	--	0.90	0.92
Crowd	0.76	0.84	0.76	0.86	0.92	0.92
QuickRotation	0.76	0.76	0.57	0.82	0.82	0.92
Parallax	0.72	0.84	0.57	0.85	0.91	0.93

**Table 3 entropy-24-01326-t003:** Comparison of stability score.

	Meshflow [1]	Liu et al. [5]	StabNet [16]	DUT [14]	Yu et al. [15]	Ours
Regular	0.84	0.81	0.84	0.84	0.89	0.87
Running	0.84	0.77	0.82	0.84	0.86	0.88
Zooming	0.74	0.85	0.81	--	0.88	0.89
Crowd	0.77	0.82	0.74	0.79	0.88	0.85
QuickRotation	0.80	0.86	0.79	0.88	0.89	0.91
Parallax	0.79	0.77	0.77	0.81	0.83	0.84

**Table 4 entropy-24-01326-t004:** Comparison of runtimes of different methods of stabilization.

Method	Runtime per Frame
MeshFlow [1]	2576 ms
Liu et al. [5]	1360 ms
DUT [14]	198 ms
Yu et al. [15]	570 ms
ours	134 ms

## Data Availability

Not applicable.

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
