# Peer review of "An Unsupervised Video Stabilization Algorithm Based on Key Point Detection"

_entropy, 2022, doi:10.3390/e24101326_

Round 1

Reviewer 1 Report

The overall paper is interesting and presents a new method for a video stabilization algorithm. I think that Introduction and Related work parts should be improved: some statements about the comparison between traditional methods and DNN-based method are not supported by argument or cited references. The method proposed by the authors is rather well described and could be improved by given more details in the Accurate alignment part for example. Experiments and results support well the conclusion of the authors.

Reviewer 2 Report

The notation is often confusing/inconsistant and typesetting is irregular. Sometimes it's M, sometimes M for the same object. v for point, but p for vector, i_2 or i2? There's some work to be done to uniformize typesetting in equations and also to name variables and constants. The attached pdf contains all the remarks.

Otherwise I think it's an interesting paper.
